# A Predictive Model for Height Tracking in an Adult Male Population in Bangladesh to Reduce Input Errors

**DOI:** 10.3390/ijerph17051806

**Published:** 2020-03-10

**Authors:** Mehdi Hasan, Fumihiko Yokota, Rafiqul Islam, Kenji Hisazumi, Akira Fukuda, Ashir Ahmed

**Affiliations:** 1Department of Advanced Information Technology, Kyushu University, Fukuoka 819-0395, Japan; nel@slrc.kyushu-u.ac.jp (K.H.); fukuda@f.ait.kyushu-u.ac.jp (A.F.); ashir@ait.kyushu-u.ac.jp (A.A.); 2Institute of Decision Science for a Sustainable Society, Kyushu University, Fukuoka 819-0395, Japan; yokota.fumihiko.785@m.kyushu-u.ac.jp; 3Medical Information Center, Kyushu University Hospital, Fukuoka 812-8582, Japan; rimaruf@med.kyushu-u.ac.jp

**Keywords:** clinical growth pattern, portable health clinic, remote healthcare, eHealth

## Abstract

The advancement of ICT and affordability of medical sensors enable healthcare data to be obtained remotely. Remote healthcare data is erroneous in nature. Detection of errors for remote healthcare data has not been significantly studied. This research aims to design and develop a software system to detect and reduce such healthcare data errors. Enormous research efforts produced error detection algorithms, however, the detection is done at the server side after a substantial amount of data is archived. Errors can be efficiently reduced if the suspicious data can be detected at the source. We took the approach to predict acceptable range of anthropometric data of each patient. We analyzed 40,391 records to monitor the growth patterns. We plotted the anthropometric items e.g., Height, Weight, BMI, Waist and Hip size for males and females. The plots show some patterns based on different age groups. This paper reports one parameter, height of males. We found three groups that can be classified with similar growth patterns: Age group 20–49, no significant change; Age group 50–64, slightly decremented pattern; and Age group 65–100, a drastic height loss. The acceptable range can change over time. The system estimates the updated trend from new health records.

## 1. Introduction

Remote healthcare is becoming popular for non-communicable diseases (such as diabetes, hypertension etc.) too. The patients need regular health monitoring and physical activity suggestions. Healthcare cost is a big burden for the individual, organizations and the government. Cost reduction for NCDs is a big agenda for all the stakeholders. Our research team has developed a remote healthcare system for this purpose and has been carrying out trials in different Asian and African countries. We call it a “Portable Health Clinic” [1,2,3] as in Figure 1.

The model of the portable health clinic is developed by considering the situation of remote areas of developing countries where the healthcare infrastructure is not sufficient e.g., insufficient or no doctors, and no clinics. There are four major components in the model [4] (a) A portable health clinic box with all the medical sensors (b) a healthcare worker who takes the measurements (BMI, blood pressure, blood glucose etc.) and sends the data to the remote doctor (c) a data server, where all the health records are archived and processed, and (d) a doctor call center, where doctors can monitor the data online and provide prescriptions.

Remote healthcare system carries erroneous data. A wrong data leads to wrong clinical decision to be made by the remote doctor. It is also very complicated and expensive process to re-examine the error data for remote healthcare systems where the doctor checks the data after all the clinical measurements are measured and digitized. Therefore, we propose a tool inside the portable health clinic software system that will detect and fix the suspicious data during measurement at the source before the data travels to the doctor.

There are two ways to detect the suspicious data- (a) at the earlier stage e.g., source during the measurements (b) at the server where all the data is archived. Most of the current approaches [5,6,7,8] to detect errors, outliers, suspicious data are based on (b). We are taking the approach of (a).

Our approach predicts an acceptance range for each individual before the healthcare worker takes the clinical measurements. The prediction is based on previous healthcare records of the individual, if it is available. If the data is not available, the system predicts for a similar situation e.g., similar age group, gender, geographical location etc.

It is not very easy to predict all the clinical measurements. We started with the anthropometric data (Height, Weight, BMI, Waist and Hip) as these are visible and has known biological growth pattern. The growth patterns vary on age and gender. Based on these logics, and past health records of the same gender and age group, we develop a predicted value for each individual. The predicted value can stay at the source where data is collected. Any data out of this range will be marked as suspicious data and the healthcare worker can re-examine and fix it immediately. This paper reports the growth characteristics of height for male. The same analysis method can be used for predicting other values too.

Portable Health Clinic archived 40,391 records for its past trials for different purposes. We found this data very resourceful to check the growth characteristics.

Understanding human growth pattern is not a new agenda of research. Many developed countries have their population growth pattern measured and archived. It is an expensive process and many developing countries could not afford to do that for each individuals or on a large sample population.

The popular clinical growth pattern metrics introduced by CDC [9] is extensively used. It demonstrates the charts by comprising a sequence of percentile curves that illustrate the dissemination of selected body measurements associated to the age of boys and girls. However, these CDC growth charts and their various derivatives addresses 2–20 years of age (Figure 2 and Figure 3). However, there are few limitations with CDC charts.

**Continuous data collection mechanism:** Data is not continuously collected. Recent eHealth data can collect data from remote clinics and or individual.**Age level:** CDC only reports clinical growth pattern of boys and girls until the age of 20. Growth pattern after the age of 20 is not available. This was due to the fact that there was no interest from the research community to investigate the similar set of characteristics of human growth pattern for ages more than 20. Human height generally stops growing at the of 20–22, however, other anthropometrics parameters (Weight, BMI, Hip and Waist) may change with age.**Anthropometric parameters:** CDC reports only height, weight, BMI and other growth parameters for children. However, recent NCD related diseases require hip, waist size to be measured. These parameters are not considered. Recent focus on non-communicable diseases (NCD) has influence on obesity of people, thus the growth characteristics of these parameters became important. In this paper, we investigate to understand if there are any significant clinical growth patterns, specifically regarding height for male over 20 years of age.

CDC is a large organization to measure, archive, analyze and report the growth pattern. It is easy to detect and remove outliers when the data is big. We assume that the measurement were taken by professionals, digitized, rechecked by professionals which minimized the measurements and digitization errors.

This model will not fit for the developing countries. Advancement of ICT, opens an opportunity to measure it by local healthcare workers. However, it carries a risk of data errors caused by the healthcare workers when they digitize the data [10].

The total number of records that we analyzed is *N* = 13,932 for male. However, this data has not been collected from the same location or from the same persons. Therefore, personal growth pattern cannot be observed. On the other hand, the data was not collected from the same location. Therefore, we cannot observe the geo-location characteristics from the data. As mentioned above, we worked on very dirty data randomly collected from different locations in Bangladesh since 2010, in different time. The other weakness of this data is that the archive does not have many follow up patients. In order to find the growth pattern, we followed a statistical process mentioned in Section 3.

We have an experimental environment to monitor NCD status in different locations in Bangladesh. We developed a Portable Health Clinic (PHC) system to collect healthcare data from remote areas. The overall system integrated in a briefcase and is carried by a healthcare worker to unreached communities. Healthcare data has been sent to a tablet triage server that identifies the level morbidity. Individual’s result is displayed as a color-coded risk category-green (safe), yellow (caution), orange (affected) and red (emergent). Orange and Red subjects are the patients and their results trigger an immediate telehealth consultancy with a remote doctor sitting in the urban area. In order to synchronized master triage server, the remote doctor make a diagnosis and sends the patient an e- prescription. We call this system the ’Portable Health Clinic’ [11]. We collected 40,391 (as of June 2018) records from different areas of Bangladesh since 2010.

The collected data has become a precious source to explore and obtain medical science related knowledge. Other researchers from our team have been using the data for the following purposes. Nazmul et al. analyzed for finding consumer behavior pattern [12,13], measuring the influences on which the consumer select to use a remote healthcare system [14]. Nohara et al. [15] analyzed the data to reduce healthcare measurement cost by excluding less important measurements. A personal health book was introduced by Seddiq et al. [16] to bring interoperability of personal healthcare data. Collecting data from remote area is always challenging. It was observed that 12% errors are made by the healthcare assistants when the measured data is manually inserted to the system [17]. Other source of errors are presented in [10]. Eiko et al. analyzed PHC data and got the predict accuracy 76.24% with inquiry and vital data, and 82.55% with adding chief complaints data [18]. In healthcare systems, an interesting topic is homecare assistance by telemedicine [18,19,20]. Some certified medical devices are tested on patients at home; parameters, and a sphygmomanometer device. the experimental devices are an ECG (electrocardiogram) device, a spirometer, an infrared Building maintenance management module: this module is integrated into the resource thermometer, a pulse oximeter, a device for hematological analysis, a monitoring device of management platform and is suitable to plan the maintenance of the building structure thus multiple parameters, and a sphygmomanometer device [21]. Tanvir et al. worked to visualize the healthcare data by defining illness and wellness score [22]. One of the important aspects of portable health clinic system was monitored to measure the morbidity of people. PHC service conducted two phases to the analysis: the first phase identified the intensity of morbidity and the second phase re-examined the morbid patients, two months later. The results show a decrease in patients to identify as morbid among those who participated in telemedicine process [23].

The data has never been studied to explore the clinical growth patterns. We demonstrated the growth pattern for height of male. Other parameters can be demonstrated in the same manner. We successfully marked two cut off points to classify three different groups with similar characteristics.

It needs to clarify that this growth pattern does not show the growth of the same person over time. One way to explain the pattern is that the height shown today is their height stopped when they were around 20 years old. We can assume the height of a male at age level 50 today is showing his height when he was 20. The height decline after the age of 49 can explain the scenario.

The rest of the paper is structured as follows: Section 2 describes the motivation and objective of this research, Section 3 explains the data pre-processing and analysis methodology to investigate the growth pattern, Section 4 demonstrates the patterns obtained from the analysis followed by a discussion of the findings and finally we conclude our research at Section 5.

## 2. About Portable Health Clinic Data and our Research Motivation

In this section, we explain the healthcare data collection mechanism for better understanding about the collected data. Figure 1 shows the healthcare data collection tool box, we call it Portable Health Clinic Box.

The PHC system contains four principal components: a back-end data server, a medical call center, and an affordable front-end portable medical case consisting medical sensors and measuring equipment, which are used to identify non-communicable diseases. The briefcase system also includes a tablet computer installed with the android application. And it can store and share a wide range of remotely gathered health care data with physicians at major health care centers. Telehealth workflow is an important attribute after PHC compactness. The healthcare worker follows a repeatable sequence of events to ensure an efficient patient turnaround for group health check-up. When screening a queue of people, the most efficient workflow is Figure 4:

**Registration:** Patients supply vital information e.g., name, age, sex, location and disease complaints etc. A data entry operator inputs data into the database (GramHealth). A patient ID is issued for follow up observations.**Index:** A set of physical check-up is performed by the healthcare assistant, the clinical data is shared with GramHealth database. The data can be uploaded automatically if the sensor have a wireless interface and is connected with the local server (the tablet PC), or the healthcare assistant manually inserts data by using an android app attached to the PHC box.**Criterion:** Patients are classified in four groups with color-coded risk stratification: green (healthy), yellow (caution), orange (affected) and red (emergency). The ‘green‘ and ’yellow’ patients are given a healthy guidance booklet. The ‘orange‘ and ‘red‘ marked patients consults with a call center doctor.**Causality:** The patients who have been marked by ‘yellow‘ and ‘red‘ should consult with the remote doctor for further investigations of their disease and explanation of their medical records. Telehealth consultancy is performed over voice and video.**Prescription and Suggestion:** At the end, remote doctor identifies disease after checking the clinical data, discussing with the patient for their symptoms analysis and his/her past health records (if any). And the doctor fills up the prescription and uploads to the GramHealth server (The GramHealth server is maintained by Grameen Communications, Bangladesh. They are operating PHC and achieving the health records. Kyushu University has a joint research collaboration agreement with them. All the privacy data has been removed and only Age, Height, Weight, BMI, Waist and Hip data are provided. The healthcare assistant prints the prescription and handover it to the patient with necessary explanations).

This is how the data has been collected and archived at the GramHealth Server. As mentioned earlier, we have now 40,391 health records for both males and females. These records are precious to investigate the growth pattern. Although most of the data are collected from the same patient only for once. Therefore, the growth pattern for the same patient cannot be achieved from this analysis. This is out of the scope of this paper. As mentioned earlier, the CDC metric system only addresses ages 2–20 years. Apparently, no other studies show correspondingly systematic growth characteristic patterns for humans over 20 years old. Very few literatures have reported this class of growth patterns using health checkup data that has been collected from Bangladesh, particularly after the age of 20. Therefore, the objective of this study is to use a unique dataset to investigate whether there may be identifiable growth characteristics of human growth after age of 20 by 5 interval age groups and consider the feasible implications of our findings.

## 3. Steps to Demonstrate Growth Pattern from Raw Healthcare Data

A total of 40,391 health records are archived in GramHealth server. Not all the records have same length. It means some patients were interested only on partial check up and were not interested in responding to survey questions. Therefore, we eliminated the incomplete records. The records which did not have all the five anthropometric data (Height, Weight, BMI, Waist and Hip) did not qualify for our analysis. We excluded those records. The archive contained kids data too. We excluded the all the data of the participants who are below 20 years of age.

The following Figure 5 explains the steps of preprocessing the data before we applied statistical analysis.

At first, we removed the incomplete, unusual and uninterested records.

**Step-1:** Filtration: We run a statistical analysis to identify the outliers. We detected outliers from our eyeball measurements. Outliers are considered to be the data which are beyond the human acceptance range. We also removed the incomplete data. We had 40,391 records, from which we found *N* = 38,856 records belongs to 20–100 years of age.**Step-2:** Gender based records: We classified the data based on gender. Number of male was $N_{male}$ = 19,054. After remove the NA, we found $N_{male}$ = 14,085 records and finally after remove the outlier the processed records was $N_{male}$ = 13,932.**Step-3:** We demonstrated a scatter plot to check the age based height pattern for each age. In this study, we focused only on male’s height (Figure 6).**Step-4:** Male height in correspondence with age by seven difference quantiles (5th, 10th, 25th, 50th, 75th, 90th, and 95th) are plotted to understanding the distribution patterns (Figure 7).**Step-5:** All the seven quantiles have been smoothened by using the popular Loess method (Figure 8). Loess represents for locally estimated scatterplot smoothing (loess stands for locally weighted scatterplot smoothing) and is one of many non-parametric regression techniques, but arguably the most flexible. A smoothing function is a function that attempts to capture general patterns in stressor-response relationships while reducing the noise and it makes minimal assumptions about the relationships among variables. The result of a loess application is a line through the moving central tendency of the stressor-response relationship. Loess is essentially used to visually assess the relationship between two variables.**Step-6:** Group wise characteristics: We observed the characteristics by classifying the participants into 17 different groups considering 5 year’s age intervals i.e., 20–24, 25–29, ⋯, 95–99, 100-).

## 4. Data Analysis and Results Discussion

In this section, we explain the analyzed data and the obtained results.

Figure 6 shows the growth characteristics of male height based on age where x-axis represents age (year) and each dot of y-axis represents height (cm). The density of the records bigger at the young age and there are only few records after the age of 80. The data was taken naturally, whoever wanted to have their health checkup performed. Therefore, the distribution of the data is not uniform.

Figure 9 shows the frequency of the data samples. where x-axis represents age groups (year) and y-axis represents the frequency. It is more clear that the prevalence of aging community is less in the chart.

Figure 10 represents the distribution of data based on five number summary (minimum, first quartile (Q1), median, third quartile (Q3), and maximum). It also gives us an idea about the outliers (shown in circles). Median height until the age of 49 do not have much variations. The inter quartile range (IQR), the distance between Q1 and Q3 is not very wide until the age of 49. However, after the age of 49, the IQR is getting wider. As we can understand from the graph, there is no significant change in terms of height up to 49 years of age is observed. The IQR height varies from 158 to 168. However, we can observe few big waves after the age of 50. Without smoothening the curve, it is difficult to conclude any tendency from this graph. Data after 80 years of age are not representative. Only 37 records were archived for this long age gap.

In order to demonstrate the pattern, we followed the same statistical method carried out by CDC. We calculated and plotted seven different quantiles as shown in Figure 7. The quantiles are the 5th, 10th, 25th, 50th, 75th, 90th, and 95th based on age. In addition to CDC’s parameters, we added average height of each age to see the central tendency of the height over age. It is observed that the maximum height is 185cm and the minimum height is recorded at 135 cm. Therefore, all the male’s heights fall into (185–135) = 50 cm of range. Now, we want to know the range of each age.

Height from the age of 20 to 49 years, there is no significant height difference. All the quartiles and average curve looks like flat lines over age. However, after 50 years of age, the curve started bumping. As mentioned earlier, due to the insufficient sample data after the age of 80, it is difficult to grasp the real tendency of height.

At Figure 8 we applied Loess (locally estimated scatterplot smoothing) method to interpolate the data and smoothen the curve. Now these charts give us a common understanding about the growth pattern.

The 5th and the 10th quantiles show a flat pattern over the age. It means people with short height are short at each age. Growth patterns remain the same. The 25th, 50th, 75th, 90th quantiles show the similar tendencies. From our eyeball measurement, we can see that the height remains almost same until the age of 49 and slightly decline after the age of 50 to 64. After the age of 65, there is a sharp decline observed. From this observation, we can classify the male height into three groups.

For a further verification, we also display the same height data over age by grouping them by five years. Every point on X-axis aggregates five year data and measure quantiles and average and plot them in Figure 11. We observe the similar tendency but in a clearer fashion.

95th quantile shows a slow declining pattern until the age of 60 and then a sharp decline after the age of 70 to 100. On the other hand, the 5th quantile shows almost no change from the age of 20 to 100. We can not classify them into different groups from this charts. We have added an average height for each age group in the chart list. It almost overlaps with the 50th quantile. If we take a closer look at the average, we can classify them into the following three age group with common trends.

We drew two cut-off points, the first one is at age 49 and the 2nd one is at age of 64, creating three age groups. We explain characteristics of these three groups itemized below.

Age group 20–49: There is no significant change in the growth at this age level. The reason could be the following: there is no significance change because generally male stop growing at the age of 18–20. Therefore, our findings comply with the natural growth of human being.Age group 50–64: A slowly decremented pattern is observed at this age zone. There could be two reasons for the height decreases with age: (a) biologically, our bone starts shrinking after this age level (b) a person of 79 years old now, was at 20 years of old in 1960s. At that period, the average height of male was 162 cm–164 cm [24]. Their growth height stopped at the age of 20. That is why the growth pattern shows a decremental characteristics.Age group 65–100: There is a drastic height loss in this age zone. Height of people may not decrease so drastically. This is quite surprising. We assume that this pattern is not representative. In fact, there are only 37 data samples. Also, it can be assumed that the people were short at when their growth level stopped at their age of twenty which happened sixty to eighty years ago. People at that time, were generally short.

Human height has steadily increased over the past 2 centuries across the globe. Historical data on heights tends to come from soldiers (conscripts), convicted criminals, slaves and servants. It is for this reason much of the historical data focuses on men. In terms of Bangladeshi male height, the data had been recorded from 1850–1980 where the maximum and minimum range was, 163.7 cm–161.6 cm. The results with previous study and found that after 38 years the range varies from 13 cm–15 cm [25]. In our analysis, the maximum and minimum range for male height is, 178 cm–148 cm.

The tendency to become shorter occurs among all races and both sexes [26]. Height loss is related to aging changes in the bones, muscles, and joints. People typically lose almost one-half inch (about 1 cm) every 10 years after age 40. Height loss is even more rapid after age 70. An individual may lose a total of 1 to 3 inches (2.5 to 7.5 cm) in height. On the other hand, one can help prevent height loss by following a healthy diet, staying physically active, and preventing and treating bone loss.

## 5. Conclusions

In this study, we introduced a 6-step model to obtain growth characteristics of a human being from remote healthcare data.This study used remote health care data from our portable health clinic project which collected anthropometric and clinical data since the year 2010. The steps include removing incomplete and uninterested data and apply statistical tools to draw smooth curves representing the growth characteristics of different age groups.

In our case analysis, we focused on male’s height to see the growth characteristics. We followed the guideline of the popular CDC growth charts and drew the height pattern of each age from 20 to 100. CDC reports growth characteristics for only 0–20 years old boys and girls. However, growth for adults after 20 has neither been widely studied, nor reported.

Our analysis removed records with empty fields, consists outliers and out of the target age group (young patients, age > 20 years). Finally, $N_{male}$ = 13,932 records were considered. Males’ height in correspondence with age by 7 different percentiles (5th, 10th, 25th, 50th, 75th, 90th and 95th) and average to represent the growth patterns of different age groups were plotted. The obtained curves were smoothened by using loess method.

There is no sharp change until the age of 49, but after the age of 50, we observe a slight decline of height and a sharp decline after the age of 64. A very small samples were available from old people (>80 years old). The obtained growth patterns at this age level are not representative. The study will continue to collect more samples from different age levels to make the curve more accurate.

The limitation of this paper is, the total number of records that we analyzed is 13,932. However, this data has not been collected from the same location or from the same persons. Therefore, personal growth pattern cannot be observed. On the other hand, the data was not collected from the same location. Therefore, we cannot observe the geo-location characteristics from the data. The major application of the anthropometric data (e.g., height, weight, hip, waist, BMI) trend is to predict a new patient’s data. If we know only the gender and age, the system can provide a prediction of those anthropometric data. A big gap between the predicted value and the measured value will be considered as suspicious data. Therefore, the data errors can be detected and removed by the healthcare worker. This is the primary goal of this research as mentioned above.

Our next target is to demonstrate the other anthropometric items such as weight, BMI, waist and hip and also investigate the same for females and compare with those of males. Once the range for age based anthropometric data is known, it will be much easier to predict measurement errors of the patients for remote healthcare systems.

## Figures and Tables

**Figure 1 ijerph-17-01806-f001:**
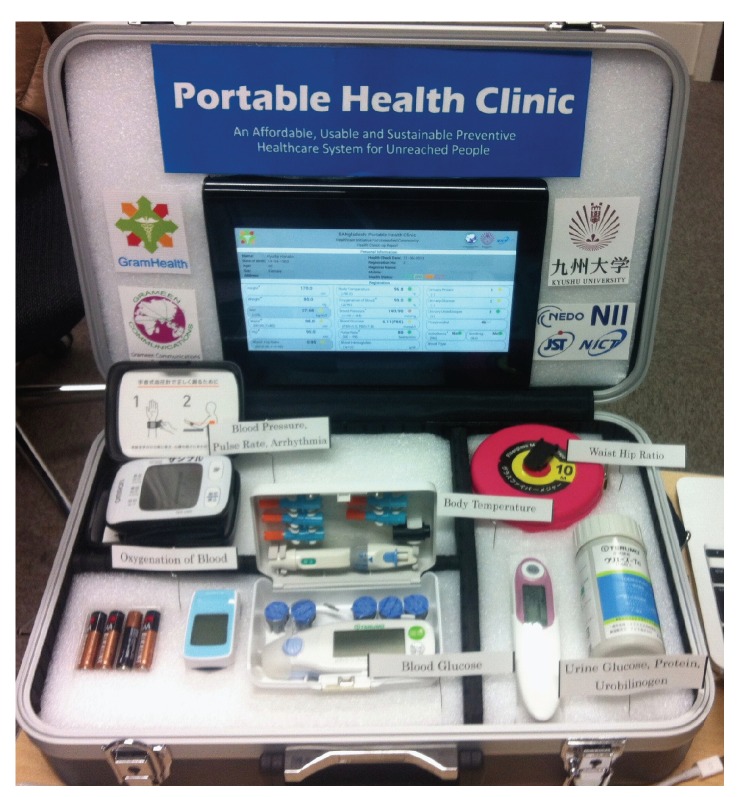
Sensors inside the Portable Health Clinic Box.

**Figure 2 ijerph-17-01806-f002:**
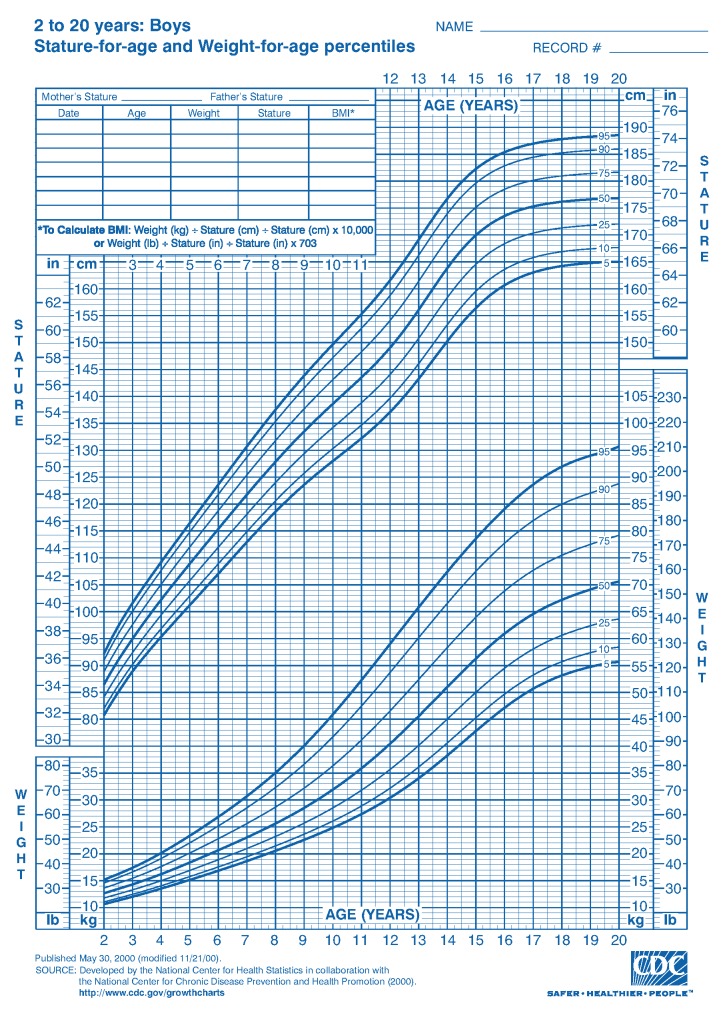
Clinical growth of boys measured by CDC (Age: 2–20 years) [2].

**Figure 3 ijerph-17-01806-f003:**
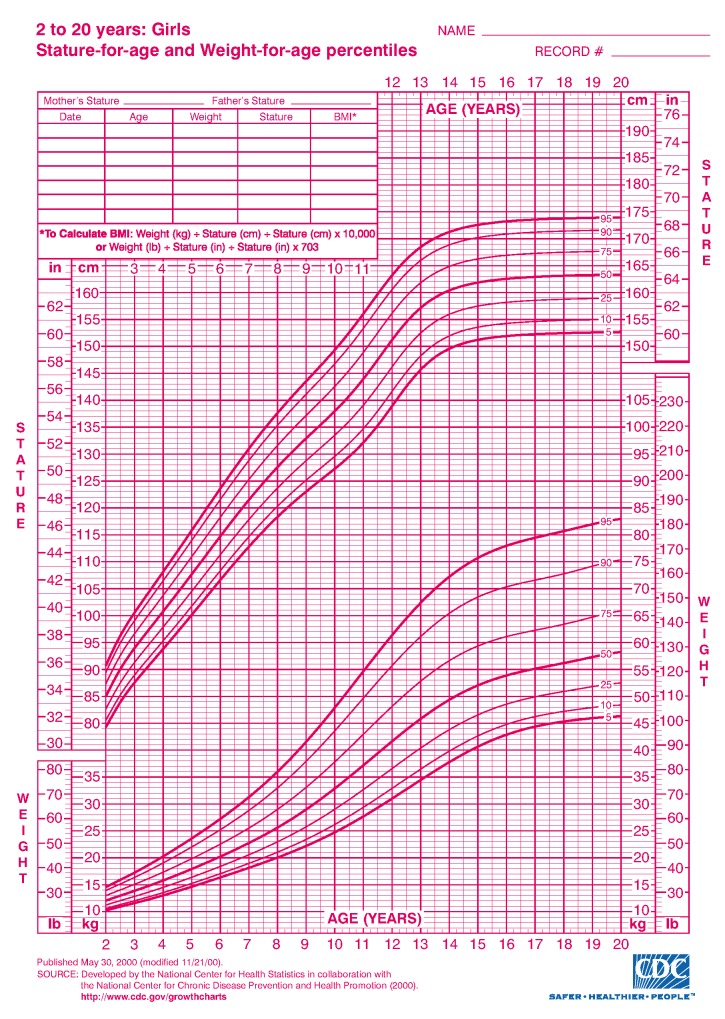
Clinical growth of girls measured by CDC (Age: 2–20 years) [2].

**Figure 4 ijerph-17-01806-f004:**
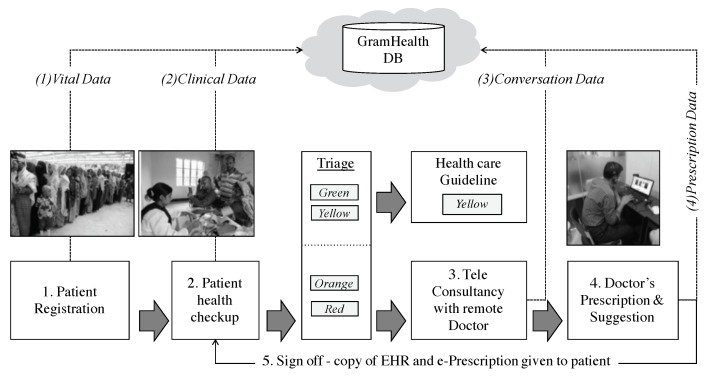
Portable Health Clinic-Data Collecting Process.

**Figure 5 ijerph-17-01806-f005:**
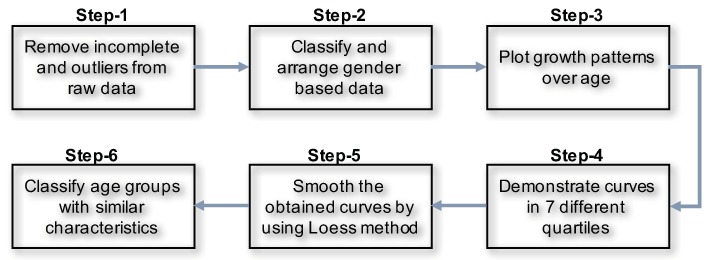
Steps to demonstrate growth pattern from remote healthcare data.

**Figure 6 ijerph-17-01806-f006:**
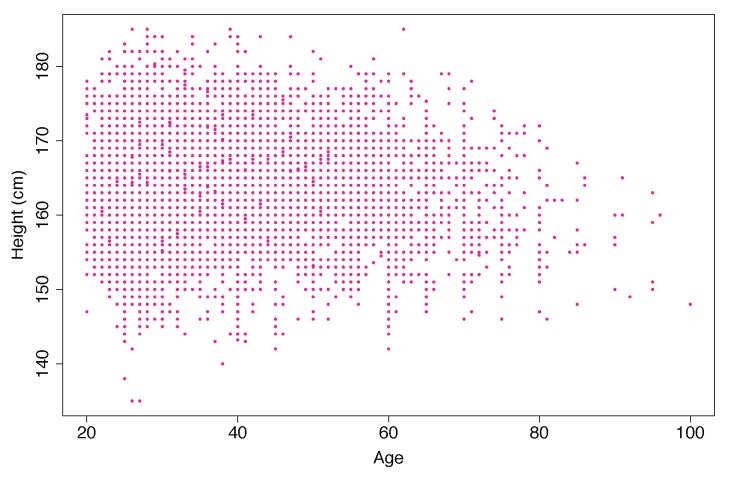
Distribution of male Height over Age providing an idea about the range of height over age and the density of the data over age.

**Figure 7 ijerph-17-01806-f007:**
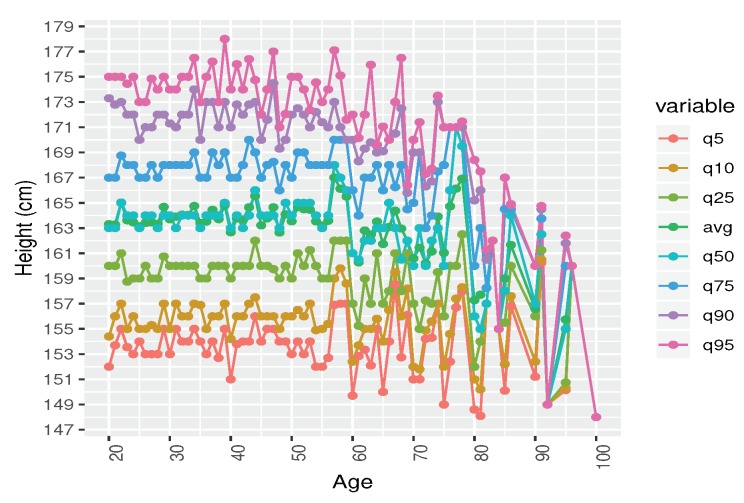
Male Height over Age in seven different quantiles: 5th, 10th, 25th, 50th, 75th, 90th, 95th and the average. The average curve almost overlaps with the 50th quantile.

**Figure 8 ijerph-17-01806-f008:**
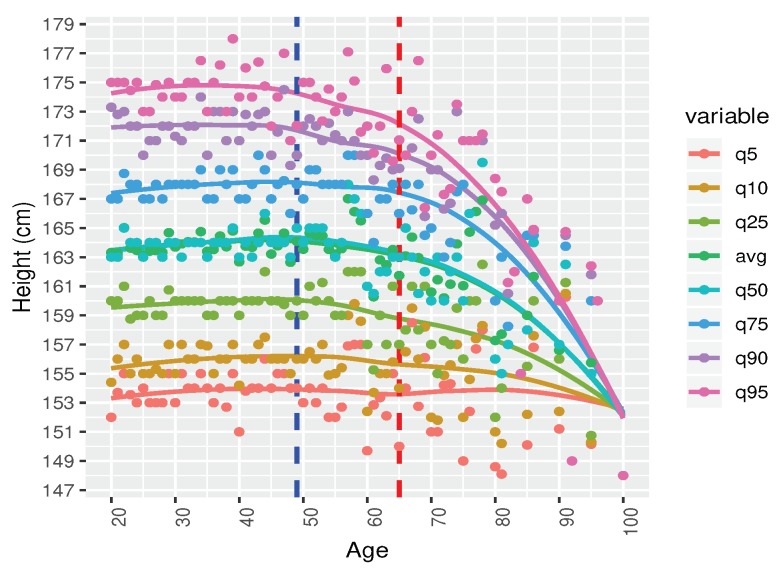
Distribution of Male Height (Age: 20–100) after applying loess method (*N* = 13,932).

**Figure 9 ijerph-17-01806-f009:**
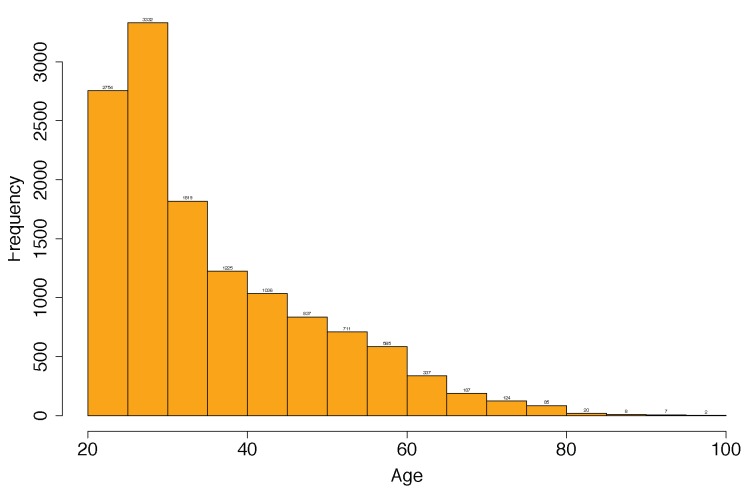
Histogram of Male Data Samples (*N* = 13,932) explaining the total number of data for each age.

**Figure 10 ijerph-17-01806-f010:**
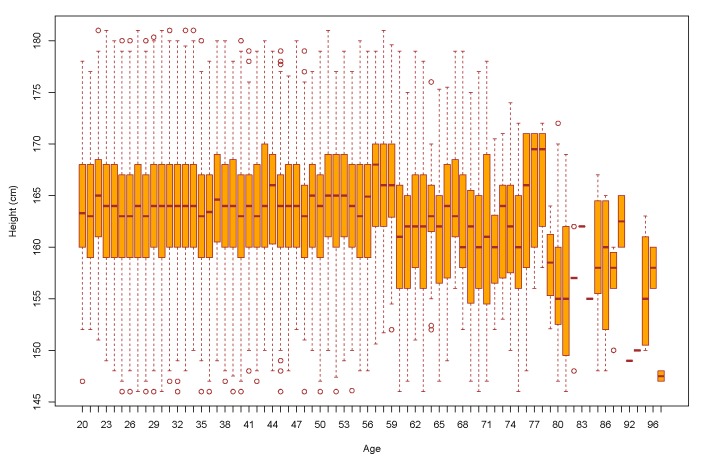
Box plot of data explaining the 1st and 3rd quantile of male height showing the range of height for each age.

**Figure 11 ijerph-17-01806-f011:**
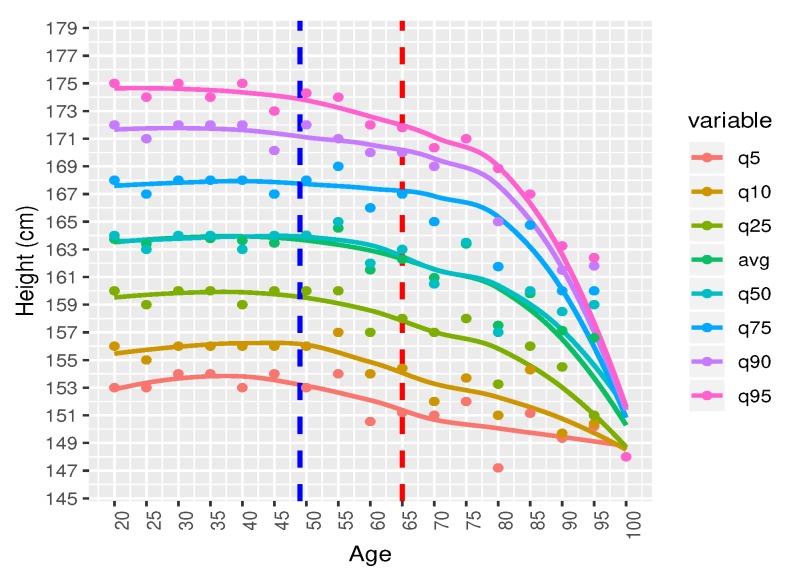
Male Height for Age 20–100: 5 years of intervals (*N* = 13,932)

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
