# Peer review of "A Predictive Model for Height Tracking in an Adult Male Population in Bangladesh to Reduce Input Errors"

_ijerph, 2020, doi:10.3390/ijerph17051806_

Round 1
Reviewer 1 Report
Machine Learning algorithms are fundamental for your research.
The data entry does not represent a high research level.
You could add for example a flowchart or cases study table embedding this topic as "design and perspectives"
Reviewer 2 Report
Authors respond point by point to reviewers' comments. However, the changes made to the text are not marked, so it is not possible to determine whether the changes improve the manuscript.
The conclusions section, for example, does not follow the standard rules for scientific publications and includes information and comments that are not conclusions, so although the authors state that they have revised their original to follow these rules, it does not appear to have been done in its entirety. The fact that it has only been applied to men is a clear limitation that must be explained properly. Females are mentioned in the summary.
Sorry, but I cannot make an assessment if the changes have not been unleashed once justified.
Reviewer 3 Report
I have carefully read the revised version of the paper submitted by the authors. Unfortunately, they did not track changes made to the text (at least in the version I could download from the website). Thus, I could not check in detail whether the provided answers comply with my remarks.
In general, the paper seems to have been revised following most of my suggestions. Some points are still pending.
The title must be changed into “A predictive model for height tracking in an adult male population in Bangladesh to reduce input errors”, as suggested by the Authors.
As I recommended, authors must always refer to the 13,932 valid records analyzed and not to the 40,391. I still see in the abstract and in various parts of the main text reference to the larger number. Please, amend the text.
My comments regarding the statistical power and the representativeness of the population (former points 9 and 10) have not been properly addressed in the discussion as potential study limitations. The authors must revise the discussion extending it to these points.
Round 2
Reviewer 1 Report
About my opinion the paper could be improved by: adding more references about the state of the art completing the full scenario of portable sensors in telemedicine and related technologies, such as:
-A cloud computing based telemedicine service. In Proceedings of the Point-of-Care Healthcare Technologies (PHT), Bangalore, India, 16–18 January 2013.
-A Study of a Health Resources Management Platform Integrating Neural Networks and DSS Telemedicine for Homecare Assistance,” Information, vol. 9, no. 176, 2018, pp. 1-20, doi:10.3390/info9070176
-Hsieh, J.; Hsu, M.W. A cloud computing based 12—Lead ECG telemedicine service. BMC Med. Inf. Dec. Mak. 2012, 12, 1–12.
Author Response
Please see the attachment.

This manuscript is a resubmission of an earlier submission. The following is a list of the peer review reports and author responses from that submission.
Round 1
Reviewer 1 Report
This study addresses an interesting (errors coding data captured in telemedicine programs) and currently relevant issue. It proposes and develops with an example a method to reduce the number of these errors. The example chosen does not relate to other measures commonly used in telemedicine programmes. There is no reference to the application of this method to blood pressure, blood glucose, etc.
The discussion does not indicate whether there are alternative methods or what measures exist to reduce errors.
Author Response
Point 1: This study addresses an interesting (errors coding data captured in telemedicine programs) and currently relevant issue. It proposes and develops with an example a method to reduce the number of these errors. The example chosen does not relate to other measures commonly used in telemedicine programmes. There is no reference to the application of this method to blood pressure, blood glucose, etc.
Response 1: Thank you for your comment. Yes, detection of errors for remote healthcare data has not been significantly studied before. This research aims to design and develop a software system to detect and reduce such healthcare data errors.
Yes, we did not consider other measures at this stage. We focused only on anthropometric items which are directly related to age. Blood pressure, blood glucose, body temperature etc. do not seem to have a relationship with age.
Point 2: The discussion does not indicate whether there are alternative methods or what measures exist to reduce errors.
Response 2: We introduced alternative methods in the introduction (line 43 to 49). Most of the alternative methods analyze data after all the data is captured. We aim to detect errors at the source. Therefore, we did not compare our results with others.

Reviewer 2 Report
In their paper, Hasan and coworkers analyzed a large sample of records from a male population in a wide age range in order to propose a predictive model of height changes tracking according to age.
The proposed approach is a predictive model based on big data and statistical modeling.
The structure of the paper does not adhere to the standards indicated by the journal https://www.mdpi.com/journal/ijerph/instructions
For instance: 4. Data analysis and Results discussion. This section is a mixture of “Methods”, “Results” and “Discussion”. There is not a structure methods section. The way the population was selected (inclusion/exclusion criteria) is not detailed.
The title does not reflect the research presented in the paper and it is too generic. I propose something like “A predictive model for height tracking in an adult Asian population”.
The authors keep writing in the manuscript that they analyzed 40,391 records, whereas they analyzed 13,932 subjects of the male population with valid data. Thus they have to replace the wrong information through the text.
The introduction is too long and must be shortened
Reference to Figure 3 (page 1, line 29) appears before that to Figure 1 and 2. The sequence of the figure must be renumbered.
Page 3, lines 85-94. This seems a discussion of data which have not yet been presented. The sentence must be deleted.
The authors describe the PHC, but they never detail how they measured body weight, height, waist circumference, etc. If the height is self- reported it might not be accurate.
Figure 7. Most of the subjects lie between 20 and 30 years. Thus, I am wondering which is the statistical power of your predictive model, particularly at an older age where few subjects were recruited.
The authors do not discuss how their sample is representative of the general population. This is a critical issue when defining a predictive model.
In general, I think it is not completely appropriate to show height according to age without putting it in relation to weight (which can affect height).
Author Response
Point 1: In their paper, Hasan and coworkers analyzed a large sample of records from a male population in a wide age range in order to propose a predictive model of height changes tracking according to age.
Response 1: Thanks for the comment.
Point 2: The proposed approach is a predictive model based on big data and statistical modeling. The structure of the paper does not adhere to the standards indicated by the journal https://www.mdpi.com/journal/ijerph/instructions. For instance: 4. Data analysis and Results discussion. This section is a mixture of “Methods”, “Results” and “Discussion”. There is not a structure methods section.
Response 2: We have restructured the paper. Section 3 explains the methodology to demonstrate the growth pattern and Section 4 describes Data Analysis, Results and Discussion.
Point 3: The way the population was selected (inclusion/exclusion criteria) is not detailed.
Response 3: This research did not select any population to collect data. We used the data, previously collected by a joint research project between Kyushu University and Grameen Communications. The population was not selected. The data was archived from all the people who wanted to receive portable health clinic services in 32 different locations in different times since 2010. We filtered the original data to find out complete records of male only. The filtration method is explained in section 3.
Point 4: The title does not reflect the research presented in the paper and it is too generic. I propose something like “A predictive model for height tracking in an adult Asian population”.
Response 4: Thank you for suggesting an attractive topic. We analyzed only Bangladeshi Male population, NOT Asian population. We tried to keep “what why and how” in the title so that the audience can easily understand. How about “A predictive model for height tracking in an adult population in Bangladesh to reduce human input errors”.
Point 5: The authors keep writing in the manuscript that they analyzed 40,391 records, whereas they analyzed 13,932 subjects of the male population with valid data. Thus, they have to replace the wrong information through the text.
Response 5: Thank you. We updated accordingly.
Point 6: The introduction is too long and must be shortened. Reference to Figure 3 (page 1, line 29) appears before that to Figure 1 and 2. The sequence of the figure must be renumbered.
Response 6: We ordered the figures.
Point 7: Page 3, lines 85-94. This seems a discussion of data which have not yet been presented. The sentence must be deleted.
Response 7: We have deleted the above lines from the text.
Point 8: The authors describe the PHC, but they never detail how they measured body weight, height, waist circumference, etc. If the height is self- reported it might not be accurate.
Response 8: Height was not self-reported. It was measured by a trained and professional healthcare worker by a standard measuring tape. Not only height, all other anthropometric and clinical data were measured by medical sensors.
Point 9: Figure 7. Most of the subjects lie between 20 and 30 years. Thus, I am wondering which is the statistical power of your predictive model, particularly at an older age where few subjects were recruited.
Response 9: Yes, that’s correct. That’s the weakest part. In fact, we have not recruited the population for this research. We analyzed the data archived by a different experiment. We do not have control of selecting the population.
Point 10: The authors do not discuss how their sample is representative of the general population. This is a critical issue when defining a predictive model. In general, I think it is not completely appropriate to show height according to age without putting it in relation to weight (which can affect height).
Response 10: We observed a correlation between weight and height. Many other papers have already reported that. We were rather interested in looking at the age level height distribution so that we can extrapolate the height of people 10, 20, 30, 40, 50 years ago. People’s height normally stops at 20 years of age. Height vs. (weight, hip, waist) will be considered in future.

Reviewer 3 Report
The Authors should apply their theory to particular algorithms such as Artificial Neural Network (or LSTM, ...) in order to prove how the decrease of input errors could improve the patient status prediction.
In this direction Authors should add a complete scenario concerning predicting algorithm in healthcare or in homecare assistance (see https://doi.org/10.3390/app9173532 or other).
Author Response
Point 1: The Authors should apply their theory to particular algorithms such as Artificial Neural Network (or LSTM, ...) in order to prove how the decrease of input errors could improve the patient status prediction. In this direction Authors should add a complete scenario concerning predicting algorithm in healthcare or in homecare assistance (see https://doi.org/10.3390/app9173532 or other).
Response 1: We have not used any machine learning algorithm in this study. This is a preliminary work to see the age-based height pattern. A healthcare worker can insert data step by step and in each step, the system will show a precaution that the measured result may be unusual. For machine learning, we will require all the data first to make a prediction. That’s why we did not consider machine learning algorithms at this stage.
